# Association of Lifestyle Changes Due to the COVID-19 Pandemic with Nutrient Intake and Physical Activity Levels during Pregnancy in Japan

**DOI:** 10.3390/nu13113799

**Published:** 2021-10-26

**Authors:** Natsuki Hori, Mie Shiraishi, Rio Harada, Yuki Kurashima

**Affiliations:** 1Department of Children’s and Women’s Health, Division of Health Sciences, Graduate School of Medicine, Osaka University, Osaka 565-0871, Japan; 25b20031@sahs.med.osaka-u.ac.jp (N.H.); harada_tea_rio@yahoo.co.jp (R.H.); y.ikss22@gmail.com (Y.K.); 2Department of Obstetrics and Gynecology, Aizenbashi Hospital, Osaka 556-0005, Japan

**Keywords:** COVID-19, lifestyle, nutrient intake, physical activity, pregnancy

## Abstract

The coronavirus disease 2019 (COVID-19) pandemic has introduced changes in our lifestyles, such as refraining from unnecessary outings. This study aimed to clarify the association of lifestyle changes due to the COVID-19 pandemic with nutrient intake and physical activity levels during pregnancy in Japan. A cross-sectional study involving 168 healthy pregnant Japanese women was conducted in 2020. Nutrient intake and physical activity levels were assessed using validated self-administered questionnaires. Participants who reported experiencing changes in both dietary habits and physical activity due to the COVID-19 pandemic were classified as the lifestyle-affected group. Analysis of covariance was used. Among primiparas, intake of the following nutrients was significantly higher in the lifestyle-affected group (n = 14) than in the unaffected group (n = 77): protein, potassium, calcium, magnesium, and vitamin B_6_. Among multiparas, the intake of dietary fiber and β-carotene were significantly lower in the lifestyle-affected group (n = 13) than in the unaffected group (n = 64). No significant differences in physical activity levels were observed in accordance with the lifestyle changes. These findings suggest that lifestyle changes due to the COVID-19 pandemic have positive effects on nutrient intake during pregnancy in primiparas, whereas in multiparas, these changes have negative effects.

## 1. Introduction

The World Health Organization (WHO) declared coronavirus disease 2019 (COVID-19) as a global pandemic in January 2020 [1]. Subsequently, new cases and deaths due to COVID-19 have been rapidly increasing worldwide. Under the pandemic situation, governments in many countries have introduced measures such as lockdowns and restrictions on everyday activities. As the number of infected people in Japan increased, the public was asked to refrain from unnecessary activities outside their homes. All commercial establishments were required to close or reduce their hours of operation, and employees of companies were encouraged to work remotely. However, the level of daily living restrictions depended on the individual’s perspective since the Japanese government’s request to refrain from going out was not enforced as a lockdown.

The COVID-19 pandemic has been reported to affect lifestyle during pregnancy, including dietary intake and physical activity [2,3,4]. In a study of American women after 8 weeks of gestation [2], nearly 60% experienced dietary changes during the COVID-19 pandemic. Approximately 17% of women reported that their diets worsened during the COVID-19 pandemic, while 42% reported an improvement. A qualitative study in India [3] reported that impaired diet quality during pregnancy was because of self-restraint in going out for shopping due to fear of COVID-19 infection. This led to a decrease in the consumption of meat, fish, and fruits [3]. Conversely, a cross-sectional study [4] of 90 Spanish women in the second and third trimesters indicated no difference in the content of meals, the number of meals, and the selection of ingredients before and during the COVID-19 pandemic, although their physical activity levels decreased significantly during the pandemic. Thus, the effects of the COVID-19 pandemic can vary depending on the country and background of the population.

In Japan, no studies have been conducted regarding lifestyle changes in pregnant women due to the COVID-19 pandemic, although the lifestyle of women in reproductive years have been reported to change [5,6]. Japanese women in the childrearing phase reported that the intake frequency of vegetables, beans, seaweed, fish, meat, and dairy products was significantly lower during the COVID-19 pandemic than before the pandemic [5]. This was probably because of the decrease in the frequency of shopping as it became less socially acceptable to go shopping with a child/children. In addition, a previous study for Japanese women found that the COVID-19 pandemic was significantly associated with decreased moderate activity and increased body weight [6]. Such lifestyle changes may also occur during pregnancy and thereby affect dietary intake and physical activity levels. 

This study aimed to clarify the association of lifestyle changes due to the COVID-19 pandemic with nutrient intake and physical activity levels during pregnancy in Japan. This study can help healthcare professionals provide lifestyle guidance to pregnant women in the event of an infectious disease pandemic.

## 2. Materials and Methods

### 2.1. Overview of the Recruitment Process and Study Design

This study used secondary data from a cross-sectional study, which was conducted among healthy Japanese women in the second and third trimesters at a perinatal medical center in Osaka, Japan, from July to December 2020. All pregnant women who met the following eligibility criteria were recruited during medical checkup at 20–26 weeks or 32–36 weeks of gestation. The inclusion criteria were singleton pregnant women aged ≥20 years and women with sufficient Japanese literacy skills. Those with diabetes, hypertension, psychological diseases, and pregnancy complications, such as gestational diabetes mellitus and hypertensive disorders of pregnancy, were excluded from the study. 

The quick response code for answering the online questionnaire was distributed directly to the participants after obtaining informed consent. The online questionnaire included questions regarding demographic characteristics, lifestyle, and psychosocial status. They were requested to answer it while waiting for an antenatal checkup or after returning home. In addition, we asked the participants to fill out a diet questionnaire at home and bring it to the hospital during their next antenatal checkup. We resolved the issue of missing data by requesting responses to our questions through email or phone within a week of reviewing the answers to the questionnaire. We reviewed the participants’ medical charts to obtain information on their pregnancies.

This study was conducted in accordance with the Declaration of Helsinki and was approved by the Institutional Review Board of the research hospital (No.19–15) and Osaka university (No.19256). All participants provided written informed consent prior to the investigation.

### 2.2. Variables and Their Measurement

#### 2.2.1. Demographic Variables

Information on the following demographic characteristics was obtained from medical charts: maternal age, parity, height, pre-pregnancy weight, and gestational period. For parity, we categorized women who had their first pregnancy as primiparas and those who had two or more pregnancies as multiparas. Pre-pregnancy body mass index (BMI) was calculated from self-reported pre-pregnancy weight and height. The participants were classified as underweight (BMI < 18.5 kg/m^2^), normal weight (18.5 ≤ BMI < 25.0 kg/m^2^), and overweight or obese (BMI ≥ 25.0 kg/m^2^) based on the WHO criteria [7]. In addition, we obtained information on the educational level, annual income, and working status from the questionnaire.

#### 2.2.2. Lifestyle Changes Due to the COVID-19 Pandemic

The impact of the COVID-19 pandemic on dietary habits and physical activity was examined by asking the following questions: “Have your dietary habits been affected by the COVID-19 pandemic?” and “Has your physical activity been affected by the COVID-19 pandemic?”. The response options were as follows: (1) very influential, (2) somewhat influential, (3) neither, (4) very little effect, (5) no effect. The first two options were classified as affecting dietary habits or physical activity during the COVID-19 pandemic, and (3–5) were classified as not having an effect. The participants who answered that both dietary habits and physical activity were affected were classified as the “lifestyle-affected” group. Participants who responded otherwise were the “unaffected” group.

#### 2.2.3. Nutrient Intake

Nutrient intake was assessed using a validated self-administered diet history questionnaire (DHQ) [8,9,10]. The DHQ was designed to assess dietary intake over the preceding month in the Japanese adult population [8,11], which was validated for certain nutrients in pregnant women [12,13,14]. In addition, DHQ has been validated using biological markers [12,13,14,15,16]. The DHQ is a 22-page structured questionnaire that measures the daily intake of 150 foods and selected nutrients. Eight eating frequency responses are listed, ranging from “more than twice per day” to “almost never”. The following five portion size responses were listed: less than half of the standard portion size, 0.7–0.8 times the standard portion size, standard portion size, 1.2–1.3 times the standard portion size, and more than 1.5 times the standard portion size. Items and portion sizes were derived from primary data obtained from the National Nutrition Survey of Japan and from various Japanese recipe books for Japanese dishes [8]. 

We excluded participants who had possibly under-reported or over-reported, based on estimated energy requirements (EER) calculated from physical activity levels (physical activity level I: low, level II: medium, level III: high). If the energy intake calculated from the DHQ was "less than 0.5 times the level I EER and more than 1.5 times the level II EER,” participants were excluded from the analysis. In addition, nutrient intake was energy-adjusted using the density method to reduce the effect of misreported data [17], that is, nutrient densities were expressed as a ratio of nutrient and energy intake.

#### 2.2.4. Physical Activity

Physical activity level was measured using the Japanese version of the Pregnancy Physical Activity Questionnaire (PPAQ-J) [18]. The original PPAQ consists of 32 questions to assess the time spent participating in physical activity over the past month [19]. The PPAQ-J consists of the following 33 activities because one extra item was included based on the cultural background: household or caregiving (13 activities), occupational (5 activities), exercise or sports (8 activities), transportation (4 activities), and inactivity (3 activities). The intensity of each activity was based on the activity codes and metabolic equivalent (MET) values list of physical activity [20]. Each activity was classified into 4 intensities as in the original version: sedentary (<1.5 METs), light (1.5 to <3.0 METs), moderate (3.0 to <6.0 METs), and vigorous (≥6.0 METs).

#### 2.2.5. Psychosocial Variables

We assessed prenatal attachment to the fetus because it has been reported to be associated with iron intake in a Japanese study [21], which showed that a stronger attachment to the fetus correlates with a higher iron intake [21]. Prenatal attachment was examined using the Japanese version of the Prenatal Attachment Inventory (J-PAI), which has been validated [22,23]. The J-PAI consists of 21 items and measures the behavior and feelings of an expectant mother toward the fetus. Each item has 4 Likert-type responses and the total score ranges from 21 to 84. A higher score indicates a stronger attachment to the fetus. The Cronbach’s alpha value of J-PAI was 0.909 in this study.

### 2.3. Statistical Analyses

Student’s *t*-test, chi-square test, or Fisher’s exact test was performed to compare the differences in the characteristics between the lifestyle-affected and unaffected groups, between primiparas and multiparas, and between women included in and excluded from the analysis.

One-way analysis of variance (ANOVA) was used to examine the differences in nutrient intake and physical activity levels between the lifestyle-affected and unaffected groups. In addition, we used analysis of covariance to examine the differences in nutrient intake and physical activity levels between the two groups after controlling for covariates. The analyses were conducted by parity. As covariates, we considered variables such as pre-pregnancy BMI, working status, and J-PAI score, which were reported to be associated with nutrient intake or physical activity levels during pregnancy in previous studies [20,24,25]. In addition, variables strongly associated with nutrient intake or physical activity levels in the univariate analysis were added to the covariates.

All statistical analyses were conducted using IBM SPSS Statistics for Windows, version 27.0, (IBM, Tokyo, Japan). All statistical tests were two-sided, and *p* values < 0.05 were considered statistically significant.

## 3. Results

Of the 226 pregnant women (124 primiparas and 102 multiparas) recruited in the present study, 197 (107 primiparas (86.3%) and 90 multiparas (88.2%)) provided written informed consent. Responses to the questionnaire were obtained from 99 primiparas (79.8%) and 84 multiparas (82.4%), excluding those who withdrew their consent. Of them, eight primiparas and seven multiparas were excluded from the analysis due to the following reasons: unrealistic under- or over-reporting of dietary intake (six primiparas and seven multiparas), non-Japanese (one primipara), and the threatened premature delivery (one primipara). Finally, 91 primiparas (73.4%) and 77 multiparas (75.5%) were included in the analyses (Figure 1).

The characteristics of primipara and multipara participants are shown in Table 1 and Table 2, respectively. Among the primiparas, 14 women (15.4%) answered that their lifestyle was affected by the COVID-19 pandemic. There were significant differences in age (*p* = 0.018) and education level (*p* = 0.023) between the lifestyle-affected group and the unaffected group in primiparas. Among the multiparas, 13 women (20.3%) answered that their lifestyle was affected by the COVID-19 pandemic. No significant differences were observed in any of the characteristics in multiparas. There was no difference in demographic characteristics between those included in the analysis and those excluded (data not shown).

The differences in nutrient intake along with lifestyle changes among the primiparas, due to the COVID-19 pandemic, are shown in Table 3, and those among the multiparas are shown in Table 4. In primiparas, energy-adjusted intake of protein (*p* = 0.028), potassium (*p* = 0.005), calcium (*p* = 0.003), magnesium (*p* = 0.002), and vitamin B_6_ (*p* = 0.012) were significantly higher in the lifestyle-affected group than in the unaffected group. Conversely, in multiparas, energy-adjusted intake of total dietary fiber (*p* = 0.014) and β-carotene (*p* = 0.038) were significantly lower in the lifestyle-affected group than in the unaffected group. There was no significant difference in physical activity levels associated with lifestyle changes due to the COVID-19 pandemic in both primiparas and multiparas (Table 5 and Table 6). In addition, no significant differences in physical activity levels were found between women who reported that their physical activity was affected by the COVID-19 pandemic and those who reported that it was unaffected (data not shown).

## 4. Discussion

This study clarified the association between lifestyle changes due to the COVID-19 pandemic and energy-adjusted nutrient intake in pregnant Japanese women. The lifestyle-affected group among the primiparas had a significantly higher intake of some nutrients compared to the unaffected group. Contrarily, the lifestyle-affected group among the multiparas had a significantly lower intake of a few nutrients compared to the unaffected group. It is noteworthy that the COVID-19 pandemic had an opposite impact on the nutritional intake of primiparas and multiparas. Meanwhile, no significant differences in physical activity levels were observed between the lifestyle-affected and unaffected groups. In our study, only 15.4% of primiparas and 20.3% of multiparas changed their lifestyle due to the COVID-19 pandemic. The proportions were lower than expected based on previous studies [2,6,26]. One possible reason for this is that perhaps the number of patients with COVID-19 was not high during the investigation period in our survey area. Therefore, the lifestyle of most pregnant women might have been unaffected by the pandemic due to their low fear of infection.

### 4.1. Association of Lifestyle Changes Due to the COVID-19 Pandemic with Nutrient Intake

In primiparas, energy-adjusted intake of protein, potassium, calcium, magnesium, and vitamin B_6_ were significantly higher in the lifestyle-affected group. This may be because the frequency of eating at home has increased due to the COVID-19 pandemic, and awareness of the importance of a well-balanced diet has increased during the first pregnancy. The Food Education Awareness Survey [27] for women over 20 years of age in Japan showed that the frequency of cooking and eating at home has increased during the COVID-19 pandemic compared to before. Decreased frequency of eating out was reported to be associated with increased nutrient intake, including dietary fiber, vitamin C, iron, magnesium, and potassium [28]. During the first pregnancy, increased frequency of cooking and eating at home might have led to an increased intake of some essential nutrients. A qualitative study showed that Japanese primiparas were conscious of the importance of a well-balanced diet for fetal health [29]. Proper intake of nutrients during pregnancy is recommended for fetal growth and development and the prevention of pregnancy complications [30,31,32]. In general, healthcare professionals apprise pregnant women regarding the importance of nutrient intake during their prenatal checkup. Lifestyle changes due to the COVID-19 pandemic might have given primiparas the opportunity to rethink the importance of their diets.

In multiparas, energy-adjusted intake of total dietary fiber and β-carotene were significantly lower in the lifestyle-affected group than in the unaffected group; this is probably due to the difficulty involved in obtaining vegetables that contain dietary fiber and β-carotene. A Japanese study reported that the frequency of shopping was significantly reduced because of self-restraint in going out during the pandemic compared to before [33]. The frequency of shopping might be particularly affected among the multiparas because shopping with a child/children during the pandemic was seen negatively by society. In addition, self-restraint with regard to shopping may occur due to fear of infection to themselves and their child/children, as reported in other countries [3,34,35]. The reduced frequency of shopping often makes it more difficult to obtain fresh vegetables [5]. Therefore, multiparas are more likely to reduce vegetable intake during an infectious disease pandemic. The intake of dietary fiber and β-carotene is important for fetal development and gestational diabetes mellitus prevention [36,37]. However, the mean total dietary fiber intake, which was lower in the lifestyle-affected group, was far lower than the tentative dietary goals set by the Ministry of Health, Labour and Welfare in Japan. We consider that identifying pregnant women at higher risk of inadequate nutrient intake due to the event of an infectious disease pandemic will be useful for dietary guidance during pregnancy.

### 4.2. Association of Lifestyle Changes Due to the COVID-19 Pandemic with Physical Activity Levels

Physical activity levels during pregnancy did not differ regardless of lifestyle changes caused by the COVID-19 pandemic. A previous study of Japanese women revealed that physical activity levels were lower during the pandemic than before [5]. However, in another study, no significant differences in physical activity levels were found before and during the pandemic in Japanese women [38]. The reason for the contradictory results might be differences in the characteristics of the population and survey areas. Although Japan implemented measures such as social distancing and restrictions on everyday activities, they were not enforced as strictly as in other foreign countries. Therefore, based on personal characteristics of the people as well as the type of residential area, the level of adherence to preventive measures varies. The prevalence of COVID-19 was not high in our survey area during the investigation period. Therefore, the degree of impact of the COVID-19 pandemic on physical activity levels might have been small. Moreover, women who experienced lifestyle changes due to the COVID-19 pandemic, such as starting remote work from home, may try to maintain their physical activity levels by performing alternative activities. During pregnancy, most Japanese women pay attention to weight control to prevent pregnancy complications and for their body image [24,39]. Therefore, as a result of taking care to maintain physical activity levels for weight control, no difference in physical activity between the two groups could have been observed.

### 4.3. Limitations

This study had three limitations. First, self-reported assessment tools of dietary intake and physical activity often have measurement errors, such as under-reporting or over-reporting. Regarding nutrient intake, statistical analysis was performed after excluding relevant subjects based on the criteria of under-reporting and over-reporting of energy intake. However, it may not be possible to completely eliminate the effects of measurement errors. Second, the number of participants was small because of secondary data, possibly reducing the overall statistical power. Third, we need to carefully consider generalizing because the research data was obtained from one facility in Osaka. However, the mean age and education levels of our participants were almost the same as those in the national data [40,41]. In addition, the values of nutrient intake and physical activity in the unaffected group were comparable to the results of previous studies conducted before the pandemic [24,25]. Osaka is the third most populous city in Japan [42]; therefore, the results of our study are applicable to healthy Japanese women with singleton pregnancies in urban areas.

## 5. Conclusions

The proportion of pregnant Japanese women whose lifestyles were affected by the COVID-19 pandemic was only 15–20%. The lifestyle-affected group among primiparas had a significantly higher intake of important nutrients such as protein, potassium, calcium, magnesium, and vitamin B_6_. Conversely, the lifestyle-affected group among multiparas had a significantly lower intake of total dietary fiber and β-carotene. However, no association between physical activity levels and lifestyle changes due to COVID-19 was observed. Our findings are useful in estimating which pregnant women are at high risk of inadequate nutrient intake. In the event of an infectious disease pandemic, healthcare professionals would need to provide nutritional guidance to pregnant Japanese women, taking into consideration the possibility of different effects according to parity.

## Figures and Tables

**Figure 1 nutrients-13-03799-f001:**
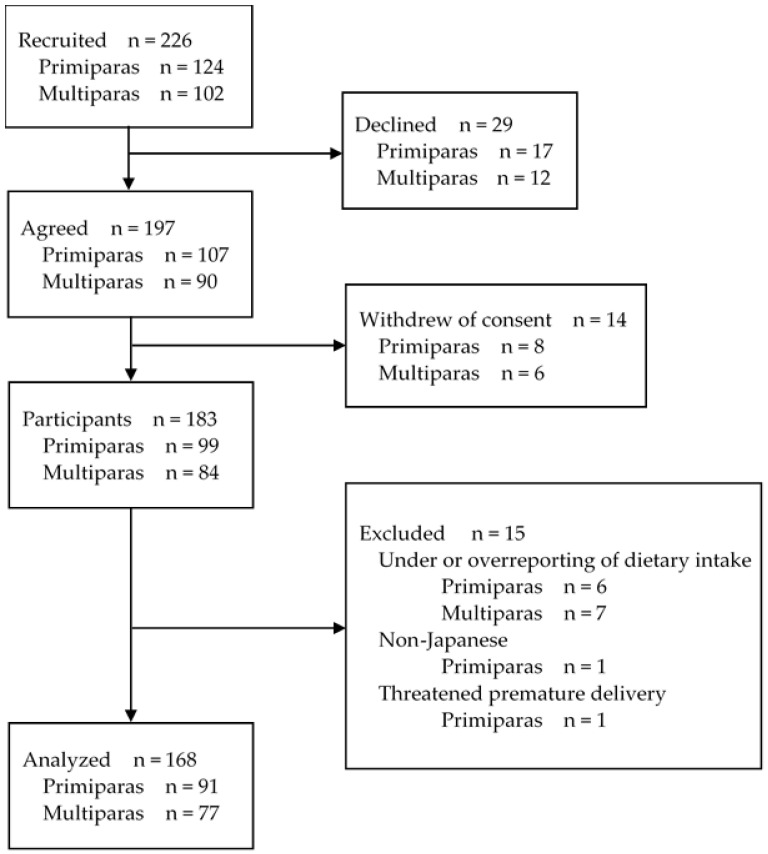
Flowchart of the participants.

**Table 1 nutrients-13-03799-t001:** The characteristics of primipara participants.

	All Participants (n = 91)			Lifestyle Changes Due to the COVID-19 Pandemic						*p*
				Lifestyle-Affected (n = 14)			Unaffected (n = 77)			
	Mean ± SD or n (%)		Range	Mean ± SD or n (%)		Range	Mean ± SD or n (%)		Range	
Age [year]	30.9	±6.1	20–45	34.4	±5.7	26–44	30.2	±6.0	20–45	0.018
Gestational period [n (%)]										
Second trimester	46	(50.5)		7	(50.0)		39	(50.6)		1.000 *
Third trimester	45	(49.5)		7	(50.0)		38	(49.4)		
Working status [n (%)]										
Working or Student	39	(42.9)		6	(42.9)		33	(42.9)		1.000 *
Housewife	52	(57.1)		8	(57.1)		44	(57.1)		
Education level [n (%)]										
University or above	22	(24.2)		3	(21.4)		19	(24.7)		0.023 *
Junior or technical college	31	(34.1)		9	(64.3)		22	(28.6)		
High school/Junior high school	38	(41.8)		2	(14.3)		36	(46.8)		
Annual income [n (%)]										
≥9 million yen	7	(7.7)		2	(14.3)		5	(6.5)		0.485 *
5–9 million yen	30	(33.0)		4	(28.6)		26	(33.8)		
0–5 million yen	54	(59.3)		8	(57.1)		46	(59.7)		
Prepregnancy BMI [n (%)]										
Underweight (BMI < 18.5 kg/m^2^)	13	(14.3)		0	(0.0)		13	(16.9)		0.247 ^†^
Normal weight (18.5–24.9 kg/m^2^)	68	(74.7)		12	(85.7)		56	(89.6)		
Overweight or obese (BMI > 24.9 kg/m^2^)	10	(11.0)		2	(14.3)		8	(10.4)		
PAI [score]	53.5	±10.8	28–77	47.4	±11.3	28–63	54.6	±10.3	35–77	0.076

BMI, body mass index; COVID-19, Coronavirus disease 2019; PAI, Prenatal Attachment Inventory; SD, standard deviation. Student’s *t*-test. * Fisher’s exact test. ^†^ Chi-square test.

**Table 2 nutrients-13-03799-t002:** The characteristics of multipara participants.

	All Participants (n = 77)			Lifestyle Changes Due to the COVID-19 Pandemic						*p*
				Lifestyle-Affected (n = 13)			Unaffected (n = 64)			
	Mean ± SD or n (%)		Range	Mean ± SD or n (%)		Range	Mean ± SD or n (%)		Range	
Age [year]	33.8	±5.1	23–47	34.6	± 5.9	26–47	33.6	±5.0	23–43	0.693
Gestational period [n (%)]										
Second trimester	40	(51.9)		4	(30.8)		36	(56.3)		0.130 *
Third trimester	37	(48.1)		9	(69.2)		28	(43.8)		
Working status [n (%)]										
Working or Student	38	(49.4)		5	(38.5)		33	(51.6)		0.545 *
Housewife	39	(50.6)		8	(61.5)		31	(48.4)		
Education level [n (%)]										
University or above	20	(26.0)		1	(7.7)		19	(29.7)		0.283 *
Junior or technical college	23	(29.9)		5	(38.5)		18	(28.1)		
High school/Junior high school	34	(44.2)		7	(53.8)		27	(42.2)		
Annual income [n (%)]										
≥9 million yen	3	(3.9)		0	(0.0)		3	(4.7)		0.216 *
5–9 million yen	34	(44.2)		3	(23.1)		31	(48.4)		
0–5 million yen	40	(51.9)		10	(76.9)		30	(46.9)		
Prepregnancy BMI [n (%)]										
Underweight (BMI < 18.5 kg/m^2^)	12	(15.6)		1	(7.7)		11	(17.2)		0.403 ^†^
Normal weight (18.5–24.9 kg/m^2^)	53	(68.8)		11	(84.6)		42	(65.6)		
Overweight or obese (BMI > 24.9 kg/m^2^)	12	(15.6)		1	(7.7)		11	(17.2)		
PAI [score]	48.3	±10.4	34–79	49.2	±9.4	37–66	48.2	±10.7	22–79	0.678

BMI, body mass index; COVID-19, Coronavirus disease 2019; PAI, Prenatal Attachment Inventory; SD, standard deviation. Student’s *t*-test. * Fisher’s exact test. ^†^ Chi-square test.

**Table 3 nutrients-13-03799-t003:** Differences in nutrient intake between the lifestyle-affected and unaffected groups in primiparas (n = 91).

		Lifestyle Changes Due to the COVID-19 Pandemic				ANOVA		ANCOVA *	
		Lifestyle-Affected (n = 14)		Unaffected (n = 77)					
	Unit	Mean	SD	Mean	SD	F	*p*	F	*p*
Energy	(kcal/day)	1675	305	1668	430	0.004	0.952	0.038	0.847
Protein	(% energy)	15	1	13	2	5.845	0.018	4.988	0.028
Fat	(% energy)	33	5	33	6	0.001	0.978	0.021	0.884
n-3 polyunsaturated fatty acid	(% energy)	1.2	0.3	1.1	0.3	0.258	0.613	0.067	0.796
n-6 polyunsaturated fatty acid	(% energy)	6.0	1.0	5.9	1.2	0.215	0.644	0.020	0.887
Eicosapentaenoic acid	(% energy)	0.06	0.03	0.05	0.04	0.408	0.524	0.138	0.711
Docosahexaenoic acid	(% energy)	0.12	0.05	0.10	0.05	0.913	0.342	0.232	0.632
Carbohydrate	(% energy)	52	6	53	7	0.376	0.541	0.136	0.713
Total dietary fiber	(g/1000 kcal)	6.6	1.5	6.3	1.8	0.388	0.535	0.744	0.391
Sodium	(mg/1000 kcal)	2120	380	2093	465	0.041	0.841	0.221	0.640
Potassium	(mg/1000 kcal)	1296	257	1132	246	5.251	0.024	8.518	0.005
Calcium	(mg/1000 kcal)	341	107	272	89	6.641	0.012	9.087	0.003
Iron	(mg/1000 kcal)	3.9	0.5	3.6	0.7	1.991	0.162	2.223	0.140
Magnesium	(mg/1000 kcal)	132	22	116	22	6.791	0.011	10.040	0.002
Zinc	(mg/1000 kcal)	4.2	0.3	4.0	0.5	2.004	0.160	1.523	0.221
Copper	(mg/1000 kcal)	0.56	0.08	0.55	0.11	0.132	0.717	0.466	0.497
Vitamin D	(μg/1000 kcal)	2.8	0.9	2.6	1.6	0.173	0.678	0.052	0.821
α-tocopherol	(mg/1000 kcal)	4.5	0.9	4.2	0.9	0.919	0.340	0.708	0.403
Vitamin B_1_	(mg/1000 kcal)	0.48	0.06	0.46	0.14	0.085	0.771	0.049	0.825
Vitamin B_2_	(mg/1000 kcal)	0.83	0.24	0.73	0.19	3.191	0.077	3.819	0.054
Vitamin B_6_	(mg/1000 kcal)	0.61	0.14	0.53	0.12	5.039	0.027	6.547	0.012
Vitamin B_12_	(μg/1000 kcal)	2.5	0.5	2.1	0.9	1.797	0.183	1.465	0.230
Folate	(μg/1000 kcal)	159	46	148	43	0.716	0.400	1.206	0.276
Vitamin C	(mg/1000 kcal)	55	28	54	25	0.016	0.900	0.001	0.975
β-carotene	(μg/1000 kcal)	1358	717	1183	609	0.920	0.340	0.916	0.342

ANOVA, analysis of variance; ANCOVA, analysis of covariance; COVID-19, Coronavirus disease 2019; SD, standard deviation. * Adjusted for gestational period (second or third trimesters), pre-pregnancy body mass index, working status, and the Japanese version of the Prenatal Attachment Inventory score.

**Table 4 nutrients-13-03799-t004:** Differences in nutrient intake between the lifestyle-affected and unaffected groups in multiparas (n = 77).

		Lifestyle Changes Due to the COVID-19 Pandemic				ANOVA		ANCOVA *	
		Lifestyle-Affected (n = 13)		Unaffected (n = 64)					
	Unit	Mean	SD	Mean	SD	F	*p*	F	*p*
Energy	(kcal/day)	1773	639	1735	405	0.076	0.784	0.000	0.995
Protein	(% energy)	14	2	14	2	0.045	0.833	0.138	0.712
Fat	(% energy)	34	5	31	6	2.500	0.118	1.033	0.313
n-3 polyunsaturated fatty acid	(% energy)	1.3	0.3	1.2	0.3	0.730	0.396	0.421	0.519
n-6 polyunsaturated fatty acid	(% energy)	6.4	0.9	5.9	1.4	1.367	0.247	0.545	0.463
Eicosapentaenoic acid	(% energy)	0.07	0.05	0.06	0.05	0.015	0.902	0.005	0.946
Docosahexaenoic acid	(% energy)	0.13	0.08	0.12	0.08	0.219	0.641	0.095	0.759
Carbohydrate	(% energy)	51	6	54	7	2.389	0.126	0.787	0.278
Total dietary fiber	(g/1000 kcal)	5.5	1.3	6.8	1.8	5.312	0.024	6.399	0.014
Sodium	(mg/1000 kcal)	2240	561	2230	727	0.002	0.961	0.111	0.740
Potassium	(mg/1000 kcal)	1064	219	1063	298	1.302	0.257	1.666	0.201
Calcium	(mg/1000 kcal)	251	75	283	116	0.952	0.332	1.653	0.203
Iron	(mg/1000 kcal)	3.6	0.8	3.8	0.9	0.494	0.484	2.054	0.157
Magnesium	(mg/1000 kcal)	114	19	119	28	0.389	0.535	1.207	0.276
Zinc	(mg/1000 kcal)	4.0	0.6	4.1	0.7	0.170	0.681	0.930	0.338
Copper	(mg/1000 kcal)	0.52	0.08	0.56	0.10	2.391	0.126	3.353	0.072
Vitamin D	(μg/1000 kcal)	2.8	1.6	2.7	1.4	0.039	0.845	0.023	0.881
α-tocopherol	(mg/1000 kcal)	4.3	0.9	4.3	1.1	0.002	0.969	0.063	0.802
Vitamin B_1_	(mg/1000 kcal)	0.44	0.09	0.46	0.10	0.281	0.598	0.957	0.332
Vitamin B_2_	(mg/1000 kcal)	0.77	0.19	0.77	0.24	0.001	0.979	0.209	0.649
Vitamin B_6_	(mg/1000 kcal)	0.52	0.14	0.54	0.13	0.242	0.625	0.350	0.556
Vitamin B_12_	(μg/1000 kcal)	2.4	1.1	2.5	1.4	0.021	0.886	0.206	0.651
Folate	(μg/1000 kcal)	143	37	158	53	0.951	0.333	1.576	0.214
Vitamin C	(mg/1000 kcal)	43	14	51	23	1.370	0.245	1.169	0.284
β-carotene	(μg/1000 kcal)	1030	396	1518	787	4.696	0.034	4.499	0.038

ANOVA, analysis of variance; ANCOVA, analysis of covariance; COVID-19, Coronavirus disease 2019; SD, standard deviation. * Adjusted for gestational period (second or third trimesters), pre-pregnancy body mass index, working status, and the Japanese version of the Prenatal Attachment Inventory score.

**Table 5 nutrients-13-03799-t005:** Differences in physical activity levels between the lifestyle-affected and unaffected groups in primiparas (n = 91).

	Lifestyle Changes Due to the COVID-19 Pandemic				ANOVA		ANCOVA *	
	Lifestyle-Affected (n = 14)		Unaffected (n = 77)					
	Mean	SD	Mean	SD	F	*p*	F	*p*
Total physical activity	34.41	15.33	30.66	14.22	1.158	0.285	1.082	0.301
Activity intensity								
Sedentary (<1.5 METs)	11.81	5.00	11.14	5.58	0.248	0.619	0.637	0.427
Light (1.5–<3.0 METs)	15.86	8.08	13.59	6.93	1.655	0.201	1.286	0.260
Moderate (3.0–<6.0 METs)	6.07	11.93	5.38	8.62	0.090	0.765	0.226	0.636
Vigorous (≧6.0 METs)	0.05	0.17	0.11	0.65	0.148	0.701	0.009	0.924
Activity type								
Household/Caregiving	6.81	2.63	6.96	3.57	0.033	0.857	0.266	0.607
Occupation	9.54	17.09	5.51	10.35	1.869	0.175	2.355	0.128
Sports/Exercise	0.96	1.1	1.41	2.26	0.782	0.379	0.889	0.348
Transportation	2.49	0.54	2.82	1.24	1.339	0.250	0.792	0.376
Inactivity	15.14	6.21	14.24	7.40	0.253	0.616	0.772	0.382

ANOVA, analysis of variance; ANCOVA, analysis of covariance; COVID-19, Coronavirus disease 2019; SD, standard deviation. * Adjusted for gestational period (second or third trimesters), pre-pregnancy body mass index, and working status.

**Table 6 nutrients-13-03799-t006:** Differences in physical activity levels between the lifestyle-affected group and unaffected group in multiparas (n = 77).

	Lifestyle Changes Due to the COVID-19 Pandemic				ANOVA		ANCOVA *	
	Lifestyle-Affected (n = 13)		Unaffected (n = 64)					
	Mean	SD	Mean	SD	F	*p*	F	*p*
Total physical activity	36.86	16.50	38.59	14.96	0.157	0.693	0.309	0.580
Activity intensity								
Sedentary (<1.5 METs)	7.86	3.36	7.46	4.53	1.347	0.148	1.990	0.317
Light (1.5–<3.0 METs)	20.10	8.36	20.02	8.31	0.001	0.975	0.002	0.964
Moderate (3.0–<6.0 METs)	11.14	9.15	11.66	9.53	0.037	0.847	0.036	0.851
Vigorous (≧6.0 METs)	0.00	0.00	0.07	0.31	0.786	0.378	1.246	0.268
Activity type								
Household/Caregiving	19.02	7.22	20.47	11.18	0.231	0.632	0.528	0.470
Occupation	7.55	11.77	6.16	8.90	0.264	0.609	0.768	0.384
Sports/Exercise	0.60	0.87	0.91	1.49	0.597	0.442	1.172	0.283
Transportation	1.28	0.45	1.44	0.68	0.542	0.466	0.221	0.641
Inactivity	7.99	5.01	9.80	6.21	0.812	0.373	0.308	0.582

ANOVA, analysis of variance; ANCOVA, analysis of covariance; COVID-19, Coronavirus disease 2019; SD, standard deviation. * Adjusted for gestational period (second or third trimesters), pre-pregnancy body mass index, and working status.

## Data Availability

Not applicable.

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
