# Peer review of "Association of Lifestyle Changes Due to the COVID-19 Pandemic with Nutrient Intake and Physical Activity Levels during Pregnancy in Japan"

_nutrients, 2021, doi:10.3390/nu13113799_

Round 1

Reviewer 1 Report

Review ofNutrients-1386114Title: Association of lifestyle changes due to the COVID-19 pandemic with nutrient intake and physical activity levels during pregnancy in Japan                                      

The authors undertook a an interesting topic about the effect of lifestyle changes due to the COVID-19 pandemic on nutrient intake and physical activity levels during pregnancy in Japan.               

The Abstract section – it's written correctly.               

The Introduction section - it's written correctly, too. However, it is not known what type of restrictions due to the pandemic were in force in Japan at that time. This is important information showing which changes in the behavior of pregnant women resulted from restrictions imposed by the state and which resulted from fear.              

The Methods section - It is not clear whether the exclusion criteria, in addition to the diseases mentioned, also included those that developed during pregnancy, such as pre-eclampsia, gestational diabetes, etc.  The authors do not state how it was verified whether the changes in eating behavior resulted from a pandemic and not from being pregnant.In line 92, the parenthesis for the reference should be square.The recruitment process for the study would be clearer if it were presented in the form of a flowchart diagram.Among the statistical analyzes, the authors mention the Pearson correlation, however, in the presented results, there are no data presenting such an analysis.Under each of the tables, there should be information about the tests with which the analysis of the results was carried out.The Results and Discussion sections are written correctly.              

 The manuscript needs minor revision.

Author Response

We appreciate a helpful suggestion. We have revised our manuscript ID nutrients-1386114 on the basis of your comments. The revision is highlighted by using “red text” in the attempted document. 

Reviewer 2 Report

This study aimed to clarify the association of lifestyle changes due to the COVID-19 pandemic with nutrient intake and physical activity levels during pregnancy in Japan classified as the lifestyle-affected groups.
The first problem with this study is that this study used secondary data from an other cross-sectional study. I wonder if the study design considered this analysis a priori as a secondary objective of the study or if it is a use of data to get more out of the study.
91 primiparas (73.4%) and 77 multiparas (75.5%) were included in the analyzes. Only 15.4% of the primiparas and 20.3% of the multiparas report changes in their life-style. With this first step, I believe that the first thing is to define whether the percentage of pregnant women who change their lifestyle is significant. I think that the percentage is very small and I suppose that it is not significant. I do not find that the authors did this statistical study.
With this, the only conclusion is that the COVID-19 pandemic did not change the life habits of Japanese pregnant women.
The rest of the statistical studies are of little interest.
Comparing means between a group of 14 versus another of 77 patients is very complicated even with non-parametric statistical tests.
In addition, they present subjective data according to a dietary or physical exercise survey, but they do not present objective data such as weight gain of the mother or fetus, obstetric complications, body composition, etc.
With all these methodological problems in addition to being a single-center, cross-sectional study, I don't think the conclusions are reliable.

Author Response

(The authors gave the same response as above.)

Round 2

Reviewer 2 Report

The article cannot be improved any more.

Author Response

Thank you for your comment.